# Effects of intermittent hypoxia on adipose-derived mesenchymal stem cells in protecting alveolar type II cells from injury

RuiYin Lai[1], SuHeng Chen[1], JunJie Cui[1], XiaoLong Liu[1], YuLan Li[2]*

1 The First School of Clinical Medicine, Lanzhou University, Gansu, Lanzhou, China, 2 The First Hospital of Lanzhou University, Gansu, Lanzhou, China

* liyul@lzu.edu.cn

## Abstract

Adipose-derived mesenchymal stem cells (ADSCs) have demonstrated significant therapeutic effects on acute lung injury. Numerous studies have reported that pre-conditioning ADSCs can enhance their therapeutic efficacy. Currently, there is a lack of research on intermittent hypoxia preconditioning of ADSCs. In this study, we subjected ADSCs to intermittent hypoxic preconditioning, followed by the detection of their expressions of HIF-1α, HGF, VEGF, and IL-10. We also evaluated the therapeutic effects of ADSCs on oxidative stress and cell apoptosis in LPS-induced MLE-12 cells, and compared the outcomes resulting from different types of hypoxic preconditioning. Finally, we concluded that Intermittent hypoxia preconditioning leads to decrease in the secretion of VEGF by ADSCs compared to sustained hypoxia pre-conditioning; but their therapeutic effects in terms of anti-apoptosis and anti-oxidation are comparable.

## 1. Introduction

Acute lung injury (ALI) is a commonly encountered clinical condition. Severe sepsis often leads to ALI, and lipopolysaccharide (LPS) induced damage to alveolar type II epithelial cells [1]. Alveolar type II cells contribute to the repair of damaged lung tissue and their damage impairs lung healing, leading to more severe injury [2,3]. Reactive oxygen species (ROS) are considered one of the shared pathways mediating apoptosis of alveolar epithelial cells [4]. Therefore, alleviating LPS-induced injury to alveolar epithelial cells is of substantial importance for the treatment of ALI.

Mesenchymal stem cells (MSCs) have become a novel therapeutic method for ALI [5]. Adipose-derived mesenchymal stem cells (ADSCs) exhibit pro-angiogenic, pro-epithelial, neurotrophic, anti-fibrotic, anti-apoptotic, and immunomodulatory properties, conferring substantial therapeutic potential [6]. The therapeutic capacity of MSCs is often attributed to their ability to release paracrine factors into the extracellular matrix that reduce tissue and cellular injury and promote healing [7]. There are evidences that condition medium

**Data availability statement:** All relevant data are within the paper and its Supporting Information files.

**Funding:** This research was supported by the National Natural Science Foundation of China (Grant No. 82460017) from YuLan Li, and cost for all antibodies and reagent kits used in this study.

**Competing interests:** The authors have declared that no competing interests exist.

(CM) derived from MSCs alleviates apoptosis and oxidative stress caused by various reasons. The conditioned medium of human amniotic stem cells can reduce the oxidative stress damage and delay the aging of $H_2O_2$ induced human dermal fibroblasts [8]. Mesenchymal cells derived from bone marrow (BMSCs) can inhibit the apoptosis of bleomycin-induced type II alveolar epithelial MLE-12 cell line by paracrine hepatocyte growth factor (HGF) and other cytokines [9]. LPS-pretreated ADSCs-CM exhibits a significant effect in inhibiting LPS-induced death of pulmonary microvascular endothelial cells [10]. MSCs-CM holds significant promise for the treatment of ALI [5,11].

It is reported that ADSCs cultured in the physiological condition (1%−5%O2) may benefit their proliferation, survival, differentiation, cytokine, trophic factors and extracellular vesicles while maintaining their stemness [12]. MSCs cultured under a 5% oxygen environment exhibit increased proliferative activity, elevated expression of cytokines and growth factors such as Interleukin-10 (IL-10), Vascular Endothelial Growth Factor (VEGF), basic fibroblast growth factor and epidermal growth factor, and accelerate wound healing [13]. Enhancing the paracrine capacity of ADSCs is an effective strategy to improve their therapeutic potential. However, different hypoxia regimens for ADSCs lead to different results, and may be suitable for different applications [14]. Many experiments suggest that intermittent hypoxia preconditioning can enhance antioxidant activity [15–17]. This novel preconditioning approach has been relatively less explored in MSCs. Studies have yielded different results regarding intermittent hypoxia in BMSCs. Specifically, research has demonstrated that intermittent hypoxia promotes adipogenic differentiation of BMSCs while reducing their osteogenic differentiation capacity [18]. In contrast, another study has revealed that intermittent hypoxia exerts a positive influence on the osteoblast differentiation of BMSCs and a negative effect on adipogenesis [19]. To date, intermittent hypoxia preconditioning of ADSCs remains unexplored. We propose that such preconditioning will produce differential effects versus sustained hypoxia, leading to improved cellular protective effects.

To verify the effects of different oxygen treatments on the therapeutic efficacy of conditioned media derived from ADSCs, we cultured ADSCs under normoxia (NO), intermittent hypoxia (IH), and sustained hypoxia (SH), and collected the corresponding conditioned media for the detection of partial soluble secretions. We then induced alveolar epithelial injury using LPS and assessed oxidative stress and apoptosis-related indicators to evaluate the effects of the conditioned media derived from ADSCs (ADSCs-CM). The therapeutic effects of ADSCs after intermittent hypoxia preconditioning were finally evaluated.

## 2. Materials and methods

### 2.1. Isolation and culture of ADSCs

The mice were purchased from the Medical Experimental Center of Lanzhou University, provided with ad libitum access to food and water, and sampled in a Good Laboratory Practice (GLP)-compliant laboratory within one week of acquisition. According to relevant ethical guidelines, the Ethics Committee of The First Hospital of Lanzhou University (LDYYLL2025−45) has approved this study. Twenty 4-week-old Specific

Pathogen Free male C57BL/6 mice were euthanized by cervical dislocation after sevoflurane anesthesia. Their inguinal subcutaneous adipose tissue was exposed and harvested. The tissue was then washed in pre-cooled Phosphate-Buffered Saline (PBS) containing 5% streptomycin-penicillin (Biosharp, Anhui, China), and any remaining blood vessels or fascia were removed. The adipose tissue was torn into small pieces and then digested with 2mL of 0.2% type I collagenase solution (Shyuanye, Shanghai, China) at 37°C for 30 minutes. After obtaining the cell suspension, cells were resuspended in complete culture medium and seeded into T25 culture flasks. The cultures were incubated at 37°C in a 5% $CO_2$ incubator. After 24 h, non-adherent cells were removed by washing with PBS. Culturing was continued until the cell confluence reached 80–90%, at which point the cells were subcultured and recorded as passage 0 (P0). The complete medium consisted of Dulbecco's Modified Eagle Medium/Nutrient Mixture F-12 (DMEM/F12; BasalMedia, Shanghai, China) supplemented with 10% fetal bovine serum (FBS; Abwbio, Shanghai, China) and without antibiotics. ADSCs at passages before P6 were used for subsequent assays.

## 2.2. Intermittent hypoxia or hypoxia treatment

ADSCs were subjected to hypoxia or intermittent hypoxia treatment in a Tri-gas Cell Culture Incubator (PH-min, Wuxi Hengyuan Biomedical Technology Co., Ltd., China). This incubator automatically regulates the oxygen concentration within the culture system based on both the preset oxygen level and the oxygen concentration in the gas cylinder, achieving the target oxygen concentration within 1 minute. When ADSCs reached 80% confluence, the spent medium was removed, and cells were washed twice with PBS before adding 3 mL of serum-free high-glucose DMEM. The intermittent hypoxia (IH) group was cultured in a tri-gas incubator (37°C, 5% $CO_2$) under cyclic conditions of 5% $O_2$ (30 min) and 20% $O_2$ (30 min) for 24 h. Controls included: a normoxia group (NO) maintained at 20% $O_2$ for 24 h, sustained hypoxia group at 5% $O_2$ for 12 h (12SH)/24 h (24SH). The conditioned medium from ADSCs was collected, filtered through a 0.22-μm syringe filter, and stored at −80°C until use.

## 2.3. LPS and ADSCs-CM treatments

LPS (MedChemExpress, Shanghai, China) was dissolved in PBS at a concentration of 1 mg/mL. MLE-12 cells (FuHeng biology, China) were treated with LPS at concentrations of 0, 1, 2.5, 5, 10, and 20 μg/mL for 24 h to establish oxidative stress-induced apoptosis models. We utilized the CCK-8 kit (Abbkine, Wuhan, China) and Western blotting to investigate LPS-induced apoptosis.

MLE-12 cells were treated with ADSCs-CM: DMEM (1:1, 5% FBS) or DMEM (5% FBS) [8] and incubated with LPS or PBS for 24 hours. Groups were named DMEM+PBS (PBS), DMEM+LPS (control) (LPS), NO-ADSCs-CM+LPS (NO), 12SH-ADSCs-CM+LPS (12SH), IH-ADSCs-CM+LPS (IH) and 24SH-ADSCs-CM+LPS (24SH).

## 2.4. Flow cytometry

FITC Conjugated AffiniPure Mouse Anti-Rabbit IgG (H+L), CD44 Antibody, CD73/NT5E Antibody (1:100, Boster Biological Technology, Wuhan, China), CD45 Monoclonal Antibody (30-F11), FITC (1:100, Thermofisher), CoraLite@ Plus 488 Anti- Mouse CD90.2 (30-H12) (1:100, Proteintech, Wuhan, China), CD34 Monoclonal Antibody(RAM34), FITC, eBioscience™, CD105 (Endoglin) Monoclonal Antibody(MJ7/18), PE, eBioscience™ (1:100, Thermofisher) were used for the identification of ADSCs.

Annexin V-EGFP/PI Apoptosis Detection kit (Abbkine, Wuhan, China), 2',7'-Dichlorodihydrofluorescein diacetate (DCFH-DA, Shyuanye, ShangHai, China) for flow cytometry. Flow cytometer (BECKMAN COULTER, U.S.).

## 2.5. Western blot

Cells in PBS, LPS, NO, 12SH, IH, 24SH groups were washed with PBS two times on ice. The cells were lysed using RIPA lysis buffer containing phosphatase inhibitors and protease inhibitors (100:1:1, Epizyme, Shanghai, China) for 20 minutes. The BCA Protein Assay kit (Boster Biological Technology, Wuhan, China) was used for protein quantification. Samples

were separated on SDS-PAGE gels (Epizyme, Shanghai, China) and transferred to polyvinylidene fluoride (PVDF) membranes. Membranes were blocked at room temperature for 20 min with Protein Free Rapid Blocking Buffer (Epizyme, Shanghai, China) and then incubated with primary antibodies:

Bax antibody and Bcl-2 antibody (1:2000, 1:1000, Wanleibio, Shenyang, China), Cleaved Caspase 3 Polyclonal antibody and Caspase 3/P17/P19 Polyclonal antibody (1:1000, 1:2000, Proteintech, Wuhan, China), Recombinant Anti-HIF-1 alpha antibody (Rabbit mAb) (1:500, Servicebio, Wuhan, China) and Anti-Beta Actin/ACTB Antibody (1:8000, Boster Biological Technology, Wuhan, China) overnight were incubated at 4°C. PVDF membranes were then incubated with HRP Conjugated AffiniPure Goat Anti-rabbit IgG (H+L) (1:10000, Boster Biological Technology, Wuhan, China) for 1 h at room temperature. Immunoreactive bands were visualized using Ultra-sensitive Multi-functional Imaging Exposure Instrument (Amersham Imager 680, GE Healthcare, U.S.). β-actin was used for an internal control. The bands were analyzed by using Image J software Version 1.54p.

## 2.6. ELISA

ADSCs-CM were collected and IL-10, HGF, VEGF levels were analyzed by ELISA (Jiangsu Meibiao Biological Technology Co., Ltd, Jiangsu, China) (Hangzhou MultiSciences Biotech Co., Ltd, Zhejiang, China). According to the manufacturer's instructions, the OD at 450 nm was determined.

## 2.7. JC-1 staining

Mitochondrial Membrane Potential Assay Kit (JC-1) (Abbkine, Wuhan, China), Photographs were taken using a high-content imaging analysis system, and the mean fluorescence intensity of the images was analyzed using Image J.

## 2.8. GSH, CCK-8 and ADSCs differentiation Kits

Reduced glutathione (GSH) assay kit (Spectrophotometric method) (NanjingjianchengBio, China). SuperKine™ Maximum Sensitivity Cell Counting Kit-8 (CCK-8) (Abbkine, Wuhan, China). Mouse Adipose-derived Mesenchymal Stem Cells Adipogenic Differentiation Medium, Mouse Adipose-derived Mesenchymal Stem Cells Osteogenic Differentiation Medium (Procell, China).

## 2.9. Statistical analysis

All assays were repeated at least three times. Multiple group comparisons were performed using one-way ANOVA, followed by post-hoc Tukey's test for all pairwise comparisons and Dunnett's test for comparisons against the control group. Pairwise comparisons were performed using the t-test. All statistical analyses were performed with Graph Pad Prism 8.0 (GraphPad Software, Inc.). Data are presented as the mean±Standard Error of the Mean (SEM). $P < 0.05$ was considered to indicate statistically significant difference.

## 3. Results

### 3.1. Identification and treatment for ADSCs

Flow cytometry analysis of cell surface markers, including CD73, CD44, CD90, CD105, CD34 and CD45, revealed that over 95% of the cells are ADSCs [20, 21] (Fig 1A). ADSCs exhibit a spindle shape and grow in a fibroblast-like pattern, adhering to the wall with a swirling arrangement (Fig 1B), and possess good adipogenic and osteogenic differentiation capabilities (Fig 1C). Fig 1D illustrates the modeling methods for hypoxia and intermittent hypoxia. After passaging, ADSCs were evenly divided into NO, 12SH, IH and 24SH groups. Once the cells reached 80% confluence, the culture medium was replaced, and the cells were transferred to different incubators for a total of 24 hours of incubation.

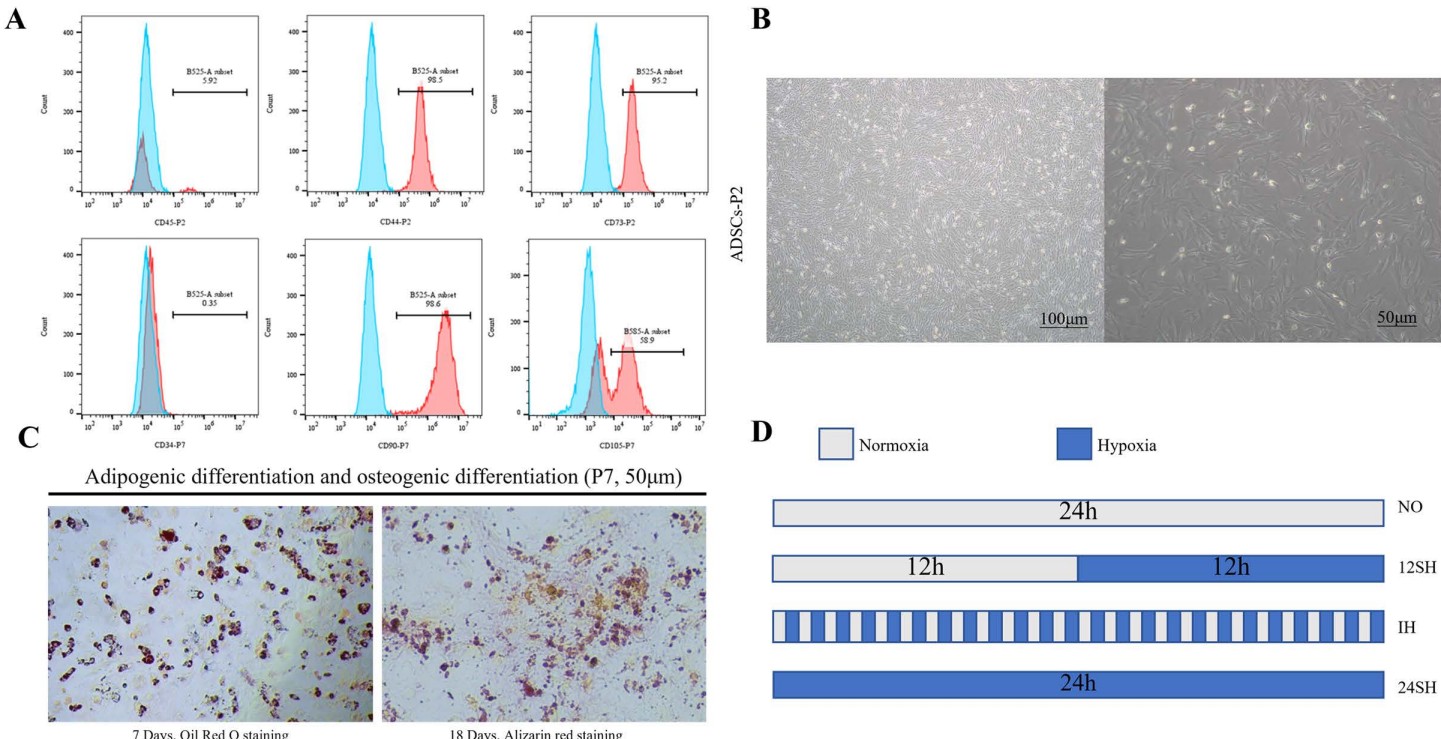

**Fig 1. Identification and treatment for ADSCs: (A)** Flow cytometry was used to identify the surface markers of ADSCs. Cells surface marker expression: CD44⁺98.5%, CD73⁺95.2%, CD45⁺5.92%, CD34⁺0.35%, CD90⁺98.6%, CD105⁺58.9%. **(B)** morphology of spindle-shaped: P2 ADSCs. **(C)** Induction of osteogenic and adipogenic differentiation of ADSCs. **(D)** The modeling methods of ADSCs.

## 3.2. Results of intermittent hypoxia treatment and selection of LPS injury concentration

Fig 2A indicates that both the 24SH group and the IH group successfully induced an increase in the expression of hypoxia-inducible factor-1α (HIF-1α). Studies have shown that the expression of HIF-1α is beneficial to the therapeutic efficacy of MSCs [22, 23]. Comparison between IH treatment and NO treatment shows no impact on ADSCs viability (Fig 2B). The ELISA results for cytokines (Fig 2C) showed that, the expression of IL-10 was significantly decreased in both the IH and the 24SH group. In the IH group and the 24SH group, cells secreted higher levels of HGF compared to other groups, while the secretion level in the 24SH group was the highest (Fig 2D). Additionally, in comparison with the NO, 12SH, and 24SH groups, IH exerted an inhibitory effect on VEGF secretion, which is different from our initial expectation (Fig 2E).

CCK-8 assay was performed to evaluate the effect of LPS on MLE-12 cell viability (Fig 2F). However, the results indicated inconsistent dose-dependent effects of LPS on MLE-12 cell viability. To overcome these challenges, we adjusted our experimental approach and employed Western Blot technology to assess Cleaved Caspase-3 expression, which confirmed that 5 μg/ml LPS induced apoptosis effectively (Fig 2G).

## 3.3. ADSCs-CM attenuates LPS-induced oxidative stress injury in MLE-12 cells

Fig 3A displays the JC-1 staining results for each group, demonstrating that LPS impaired the mitochondrial membrane potential of MLE-12 cells, while the 12SH, IH, and 24SH groups all reversed this impairment, with the IH and 24SH groups showing comparable and the most pronounced effects. Correspondingly, in Fig 3B, the ROS levels in all four ADSCs-CM treatment groups showed a significant decrease, indicating that ADSCs-CM has the ability to alleviate cellular oxidative stress. However, although the 12SH, IH, and 24SH groups all demonstrated favorable therapeutic effects in JC-1 staining

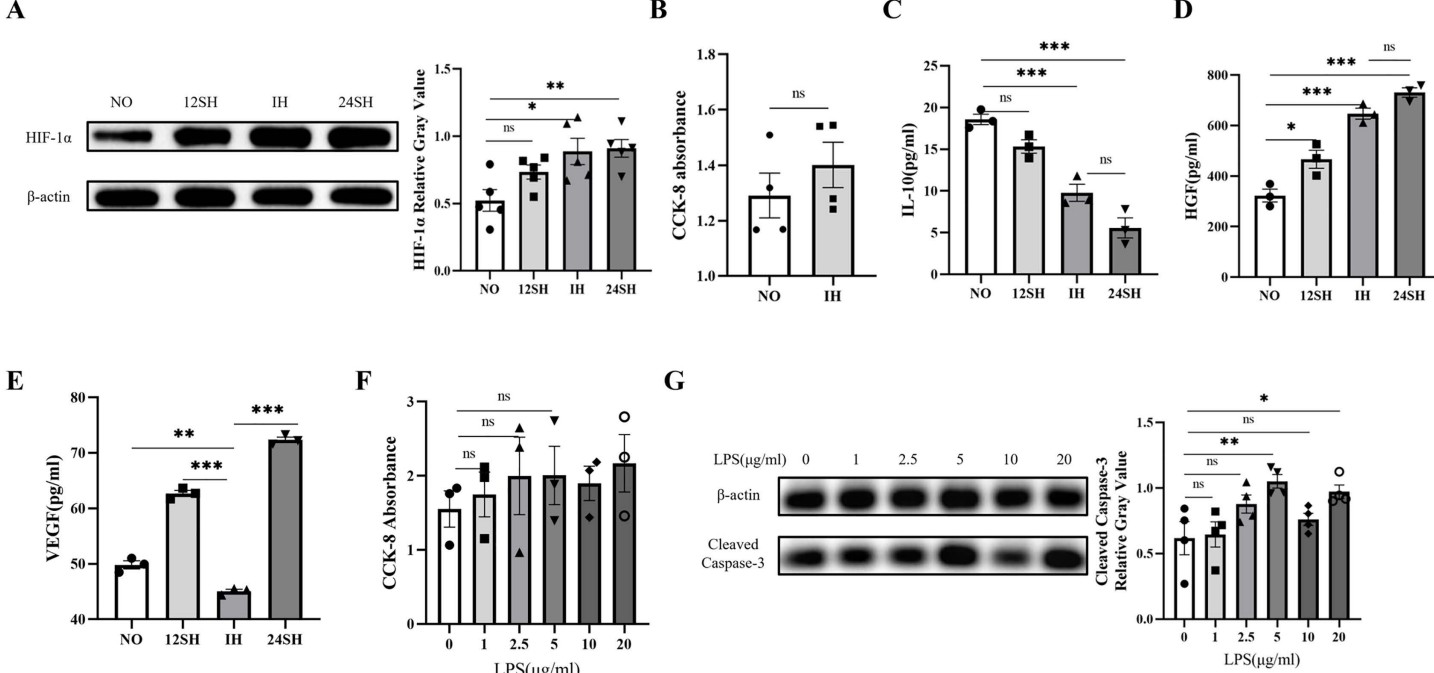

**Fig 2. Effects of Different Hypoxic Preconditioning Patterns on ADSCs and Determination of LPS-Induced Injury Concentration: (A)** Western blot images of HIF-1α in ADSCs after exposure to NO, 12SH, IH, 24SH. **(B)** CCK-8 absorbance of ADSCs after IH treatment and NO treatment. **(C)** ELISA detection of IL-10 levels. **(D)** ELISA detection of HGF levels. **(E)** ELISA detection of VEGF levels. **(F)** CCK-8 absorbance of MLE-12 cells treated with different concentrations of LPS for 24 hours. **(G)** Western blot images and statistical analysis of Cleaved Caspase-3 in MLE-12 cells after treated with LPS for 24h. mean±SEM (*P < 0.05, **P < 0.01, ***P < 0.001 control: NO group, 0 µg/ml LPS group).

and ROS flow cytometry, only the IH group exhibited a significant increase in GSH levels (Fig 3C). These results suggest the therapeutic effects of each group in LPS-induced cellular oxidative stress.

### 3.4. ADSCs-CM attenuates LPS-induced apoptosis in MLE-12 cells

LPS induced apoptosis in MLE-12 cells, which was significantly attenuated by treatment with IH-ADSCs-CM and 24SH-ADSCs-CM (Fig 4A and 4B). Consistent with this, the protein ratios of Cleaved Caspase-3/Caspase-3 and Bax/Bcl-2 were significantly decreased (Fig 4A and 4C) and the therapeutic effects of ADSCs-CM derived from the IH and 24SH groups were statistically equivalent. Interestingly, although Western blotting results indicated that neither the NO group nor the 12SH group exhibited significant effects on LPS-induced injury, flow cytometry demonstrated a reduction in apoptosis rates across all ADSCs treatment groups.

## 4. Discussion

In this study, we subjected primary murine ADSCs to intermittent hypoxia and sustained hypoxia (12h or 24h) preconditioning. Then the corresponding ADSCs-conditioned medium (ADSCs-CM) was collected, the expression of HIF-1α and the secretion profiles of IL-10, VEGF, and HGF under different hypoxic conditions were analyzed. Subsequently, we assessed the protective effects of ADSCs-CM against LPS-induced oxidative stress and apoptosis in MLE-12 cells.

Our study revealed that, compared with normoxia and 12h sustained hypoxia culture, both 24h intermittent hypoxia and 24h sustained hypoxia culture led to higher expression levels of HIF-1α and HGF, along with lower expression of IL-10 in ADSCs. When comparing 24h intermittent hypoxia with 24h sustained hypoxia, no statistically significant differences

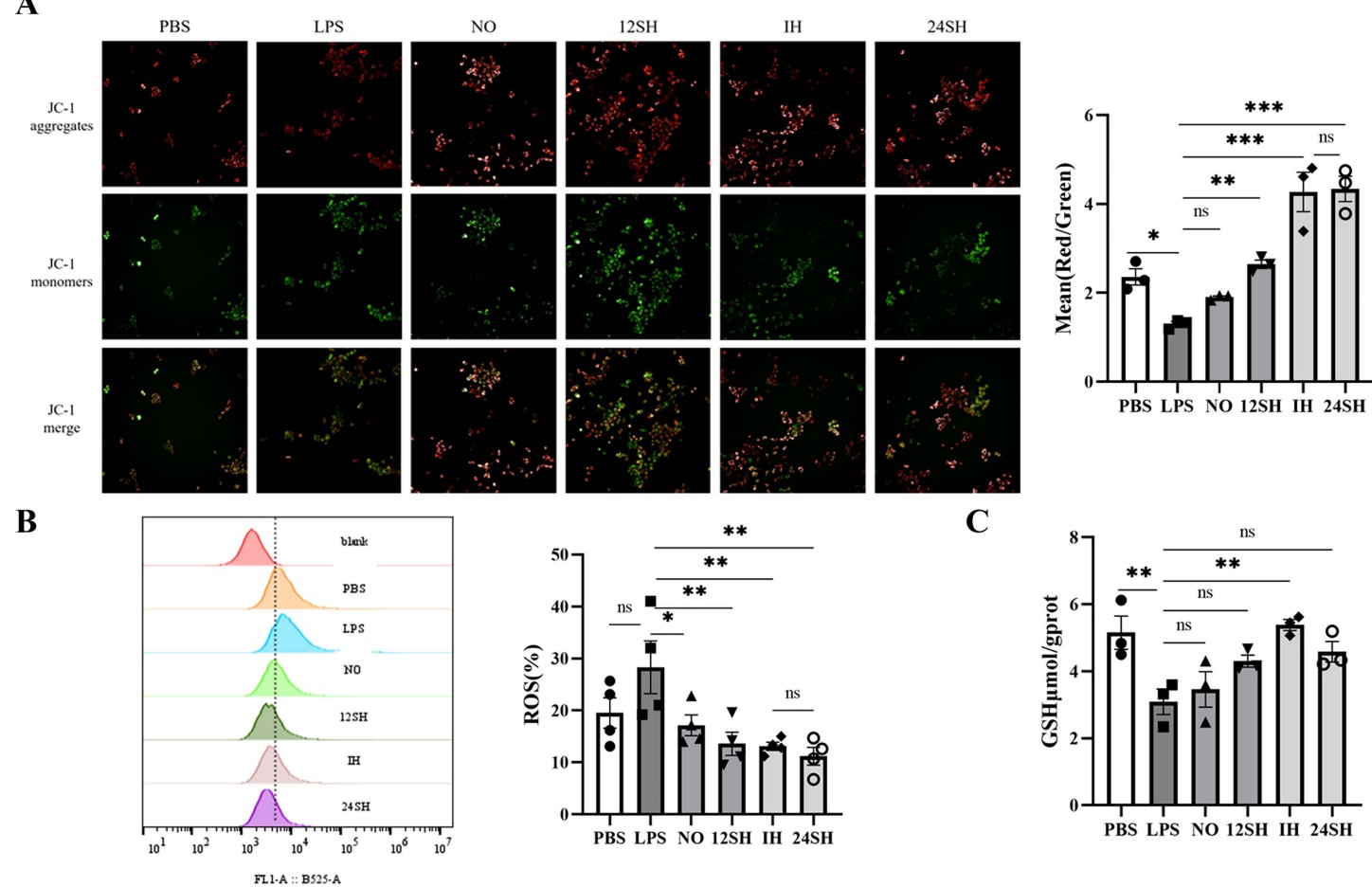

**Fig 3. LPS-Induced Oxidative Stress Injury in MLE-12 Cells and Therapeutic Effects of ADSCs-CM: (A)** JC-1 staining, the mean fluorescence intensity values were analyzed based on at least three fields of view per group. **(B)** Measurement of ROS by flow cytometry using DCFH-DA probe. **(C)** GSH (μmol/gprot). mean ± SEM (*P < 0.05, **P < 0.01, ***P < 0.001 control: LPS group).

were observed in the levels of HGF and IL-10 between the two groups. Furthermore, the intermittent hypoxia treatment demonstrated significantly reduced VEGF levels compared to all other experimental groups. In experiments aimed at treating LPS-induced alveolar epithelial cell apoptosis, ADSCs cultured under 24h intermittent hypoxia and 24h sustained hypoxia displayed stronger anti-apoptotic and antioxidant effects, with comparable therapeutic efficacy between the two conditions.

Although intermittent hypoxia and the resulting elevation of HIF is widely regarded as an injured mechanism in some research, many studies have proved its effectiveness in improving cardiovascular function, enhancing exercise and cognitive performance, facilitating high-altitude adaptation [24], and inducing cellular antioxidant responses [15]. ROS activates Nrf2, HIF-1α and other antioxidant transcription factor, then activates downstream GSH synthesis pathways and other antioxidant systems [25], which makes the preconditioning work [26].

Intermittent hypoxia preconditioning can exert cytoprotective effects like enhancing mitochondrial autophagy in hepatocytes [15] and boosting the antioxidant capacity of cardiomyocytes [27,28], thus protect them from oxidative stress injury. Intermittent hypoxia leads to more expression of HIF-1α and their target genes compared to sustained

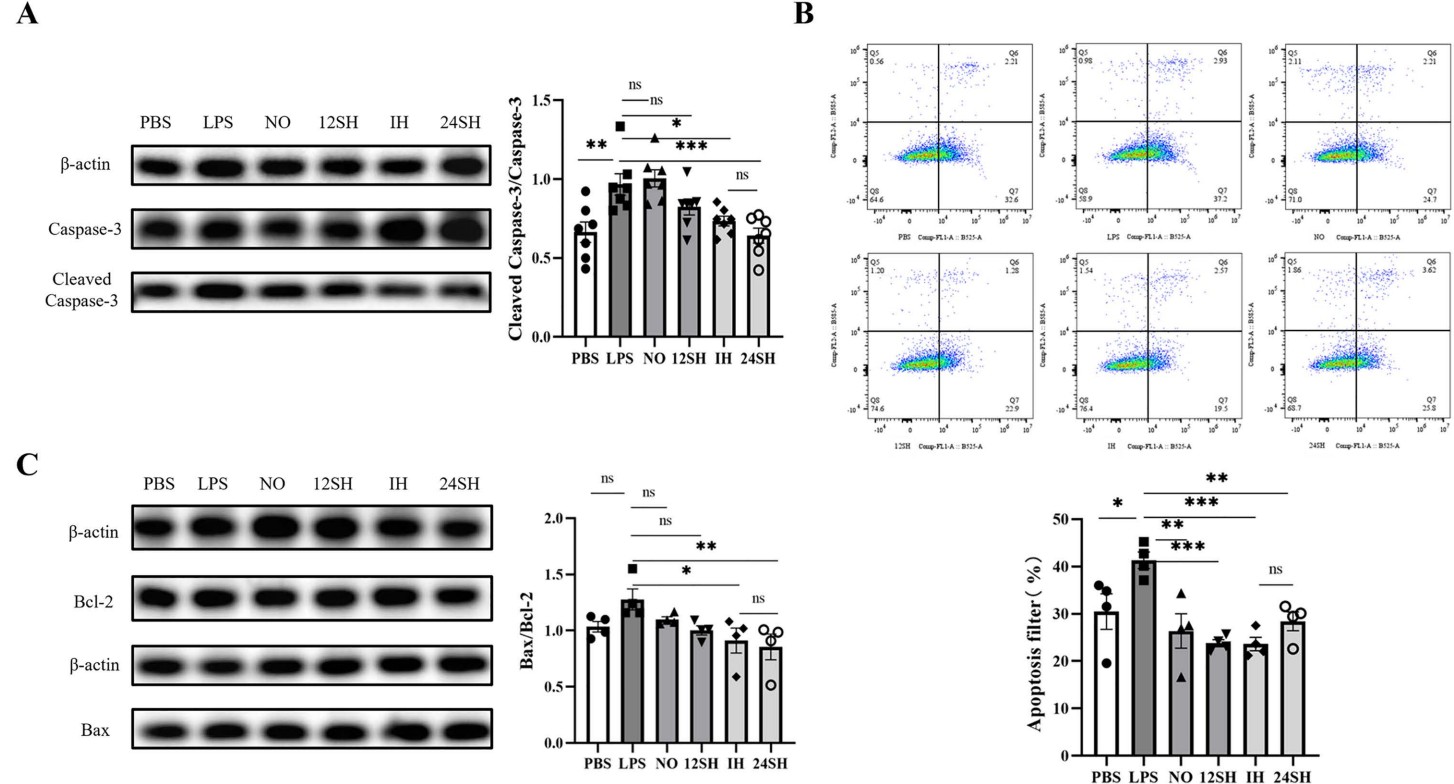

**Fig 4. LPS-Induced Apoptosis in MLE-12 Cells and Therapeutic Effects of ADSCs-CM: (A)** Western blot images and statistical analysis of Cleaved Caspase-3 and total Caspase-3. **(B)** Flow cytometry analysis of apoptosis using Annexin V-EGFP/PI staining and statistical evaluation of the results. **(C)** Western blot images and statistical analysis of Bax and Bcl-2. mean±SEM (*P<0.05, **P<0.01, ***P<0.001 control: LPS group).

hypoxia [29]. Some research suggest that intermittent hypoxia induce the phosphorylation of HIF-1α [30] and decreases the histone deacetylases enzyme activity to increase HIF-1α transcriptional activity [31]. Other research proves that BMSCs and HepG2 hepatic tumor cells exposed to intermittent hypoxia exhibit higher levels of VEGF expression compared with those exposed to sustained hypoxia [29,32]. But our research found that although IH upregulated the expression of HIF-1α, VEGF secretion was suppressed after IH preconditioning, whereas it was significantly elevated after sustained hypoxia preconditioning. It seems like the rise of HIF-1α after IH preconditioning does not lead to the expected regulation of VEGF secretion. This phenomenon might be attributed to the interference of other signaling pathways or the quick degradation of HIF-1α. During the reoxygenation phase of IH, HIF-1α undergoes rapid degradation and rise again during the hypoxic phases, continue to increase with each cycle of hypoxia [30]. This pattern of rapid HIF-1α degradation may impair its transcriptional activity and potentially affect its ability to regulate VEGF expression.

Exogenous VEGF exhibits a dual role in the context of ALI. VEGF protects alveolar epithelial cells and plays a role in repair following lung injury. However, if the alveolar capillary membrane is functionally compromised, it can lead to increased fluid flux across the exposed endothelial cells [33]. The reduction of exogenous VEGF may avoid interfering with endogenous repair signals. In this study, the decrease in VEGF in IH - ADSCs - CM did not affect its therapeutic efficacy, indicating that the specific anti – apoptotic mechanism of ADSCs - CM may have a weak association with VEGF. We believe that this decrease in VEGF may not necessarily be a negative thing, but it still requires further in vivo studies for verification.

In our study, the secretion of IL-10 by ADSCs decreased in both intermittent hypoxia treatment and sustained hypoxia treatment, which implies the immunomodulatory capacity of ADSCs declines following hypoxic preconditioning, match

what some studies have found [34–36] but contrary to some studies [37,38]. The differences in these research findings may be due to the various species sources of cells or the oxygen concentrations used in the preconditioning.

Intermittent hypoxia and sustained hypoxia can activate the adaptive capacity of cells and promote the expression of HGF, a growth factor with proliferative and anti-apoptotic effects. This capability is not exclusive to ADSCs, it also extends to other cell types, such as cardiomyocytes [39], pancreatic β cells [40], and BMSCs [41]. HGF often activates the HGF/Met/Bcl-2 signaling pathway to inhibit cell apoptosis [42]. In our experiments, we observed that after treatment with ADSCs-CM, the expression of Bax/Bcl-2 decrease, this corresponds to the increase in HGF.

We employed the JC-1 staining method to assess changes in mitochondrial membrane potential. Our results showed that ADSCs-CM exhibits a certain therapeutic effect in mitigating the LPS-induced decline in mitochondrial membrane potential. Notably, both IH-ADSCs-CM and 24SH-ADSCs-CM demonstrated superior protective effects on mitochondrial membrane potential, with comparable efficacy. This suggests that the IH-ADSCs-CM and 24SH-ADSCs-CM exhibit equivalent mitochondrial protective effects during the early stage of apoptosis. This corresponds to the flow cytometry results of ROS staining using the DCFH-DA probe. Electron leakage from the electron transport chain reacts with $O_2$ to form ROS, triggering a cascade of oxidative stress-induced apoptotic processes [43]. After treatment with IH-ADSCs-CM, the cellular GSH content increased significantly, corresponding to the activation of the cellular ROS scavenging system and the activation of cell protective mechanisms. ADSCs subjected to other preconditioning treatments exhibited no significant difference in GSH content compared to the LPS-injury control. This indicates that GSH plays a more crucial role in the therapeutic efficacy of IH-ADSCs-CM. Further research is required to determine which specific cytokine or exosome-activated pathway is responsible for the therapeutic effect that elevates cellular GSH levels.

Actually, all ADSCs-CM treatments significantly reduced intracellular ROS levels and exhibited anti-apoptotic effects in flow cytometry, but only intermittent hypoxia treatment and 24h sustained hypoxia treatment for ADSCs showed a significant downregulation in Cleaved Caspase-3/Caspase-3 and Bax/Bcl-2. This may be because flow cytometry has high sensitivity in detecting apoptotic cells and is capable of capturing early-stage apoptotic cells. The NO group and the 12SH group might have shown certain effects during the early apoptotic phase, but these effects did not persist to the time point of Western Blot detection. In flow cytometry, the excessively high proportion of early apoptotic cells obscured the overall trend. Our results demonstrate that ADSCs-CM reduces cell apoptosis, with the IH preconditioning and the 24SH preconditioning exhibiting the strongest therapeutic capabilities.

Intermittent hypoxia holds significant potential for applications in stem cell therapy, similar yet distinct outcomes induced by sustained hypoxia and intermittent hypoxia have also been observed in BMSCs[19]. The derived secretome of MSCs is not a constant mixture factors, preconditioning MSCs to enhance their therapeutic effects has been a hot research topic in recent years [44]. In this study, we used a new hypoxic preconditioning method for ADSCs and observed different effects of ADSCs-CM under varying hypoxia conditions. We used LPS to induce apoptosis in MLE-12 cells, thereby simulating the mechanistic role of LPS in acute lung injury. ADSCs represent a promising therapeutic approach for ALI [45]. We utilized ADSCs-CM to treat an in vitro model of ALI, and explored a novel preconditioning strategy for ADSCs. However, due to time and cost constraints, we did not investigate the proliferative and differentiative activities of ADSCs under IH conditions. This aspect remains to be explored in subsequent studies.

Similar with remote ischemic preconditioning, intermittent hypoxia involves exposure to appropriate and non-injurious cyclical hypoxia. Hypoxia-induced activation of ROS in tissues and cells mobilizes transcription factors to drive the production of cell-protective proteins and initiate protein kinase signaling pathways, thereby activating cellular defenses against oxidative stress, inflammation, and energy depletion [26,46]. Similar mechanisms can also be observed in sports training during high-altitude training and intermittent hypoxic training [47]. However, no study has made it clear whether intermittent hypoxia preconditioning is better than continuous hypoxia preconditioning [48]. Our research focuses on intermittent hypoxia treatment at the cellular level rather than on the whole organism. Our study has not yet clearly demonstrated whether intermittent hypoxia is superior to sustained hypoxia in terms of overall efficacy, but intermittent hypoxia preconditioning has the advantage in the GSH decrease after LPS-injury.

## 5. Conclusion

In conclusion, the adipose-derived mesenchymal stem cells preconditioned with intermittent hypoxia exhibit comparable effects to those subjected to sustained hypoxia in terms of enhancing cellular antioxidant stress and anti-apoptosis capabilities. However, it demonstrates a distinct secretory pattern in VEGF, and the therapeutic effect of the adipose-derived mesenchymal stem cells after intermittent hypoxia preconditioning holds significant effect in reversing GSH depletion induced by LPS injury. Intermittent hypoxia preconditioning holds significant research value in the field of stem cells for the therapy of acute lung injury.

## Supporting information

**S1 File. Raw Blot Data.** The raw data from Western blot.
(PDF)

**S1 Table. Minimal Dataset.** Contains key variables for analysis.
(XLSX)

## Acknowledgments

This study was technically supported by Medical Frontier Innovation Research Center of the First Hospital of Lanzhou University. We thank Dr. Jinmin Ma for expert technical support.

## Author contributions

**Conceptualization:** RuiYin Lai.

**Data curation:** RuiYin Lai.

**Formal analysis:** RuiYin Lai, SuHeng Chen.

**Funding acquisition:** Yulan Li.

**Investigation:** RuiYin Lai, JunJie Cui.

**Methodology:** RuiYin Lai, SuHeng Chen.

**Project administration:** RuiYin Lai.

**Resources:** Yulan Li.

**Supervision:** SuHeng Chen.

**Validation:** RuiYin Lai, JunJie Cui.

**Visualization:** RuiYin Lai, SuHeng Chen.

**Writing – original draft:** RuiYin Lai.

**Writing – review & editing:** XiaoLong Liu, Yulan Li.

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
