## [Decision Letter · Decision Letter 0]

3 Nov 2025

PONE-D-25-45087Effects of intermittent hypoxia on adipose-derived mesenchymal stem cells in protecting alveolar type II cells from injuryPLOS ONE

Dear Dr. Li,

Thank you for submitting your manuscript to PLOS ONE. After careful consideration, we feel that it has merit but does not fully meet PLOS ONE’s publication criteria as it currently stands. Therefore, we invite you to submit a revised version of the manuscript that addresses the points raised during the review process.

We look forward to receiving your revised manuscript.

Kind regards,

Cheng-Maw Ho

Academic Editor

PLOS ONE

2. To comply with PLOS One submissions requirements, in your Methods section, please provide additional information regarding the experiments involving animals and ensure you have included details on (1) methods of sacrifice, (2) methods of anesthesia and/or analgesia, and (3) efforts to alleviate suffering.

 [This research was supported by the National Natural Science Foundation of China (Grant No. 82460017) from YuLan Li, and cost for all antibodies and reagent kits used in this study.].

7. Your ethics statement should only appear in the Methods section of your manuscript. If your ethics statement is written in any section besides the Methods, please move it to the Methods section and delete it from any other section. Please ensure that your ethics statement is included in your manuscript, as the ethics statement entered into the online submission form will not be published alongside your manuscript.

Additional Editor Comments:

Extra work has to be done to address the reviewers' concerns.

Reviewers' comments:

Reviewer's Responses to Questions

**Comments to the Author**

1. Is the manuscript technically sound, and do the data support the conclusions?

Reviewer #1: Yes

Reviewer #2: Partly

2. Has the statistical analysis been performed appropriately and rigorously? 

Reviewer #1: Yes

Reviewer #2: Yes

3. Have the authors made all data underlying the findings in their manuscript fully available?

Reviewer #1: Yes

Reviewer #2: Yes

4. Is the manuscript presented in an intelligible fashion and written in standard English?

Reviewer #1: Yes

Reviewer #2: Yes

5. Review Comments to the Author

Reviewer #1: This study addresses an important question. The methods and results are clearly described. I have the following suggestions for the current manuscript.

The hypoxia protocols are not fully specified. For full reproducibility, please state ramp time to each O₂ setpoint. A brief note on how O₂ was monitored would be better.

Your data show similar anti-apoptotic and anti-oxidative effects for IH and 24SH. IH also shows lower VEGF and stronger GSH recovery. However, causal tests are not included. Emphasize correlation rather than superiority and add an appropriate discussion of this point in the Discussion section.

In Results 3.4, flow cytometry indicates reduced apoptosis across all CM groups, while western blots show no significant improvement with NO or 12SH. Please confirm matched assay timepoints.

In multi-panel figures (e.g., Figure 2), the order is not fully consistent. Please revise either the figure or the text so that the first in-text callouts follow alphabetical order, and update the Figure legend to match the final sequence to optimize readability.

Thank you for considering my feedback!

Reviewer #2: This paper explored the effects of intermittent hypoxic preconditioning on ADSCs. It was found that intermittent hypoxia preconditioning leads to changes in cytokine secretion compared to sustained hypoxia preconditioning. Their therapeutic effects of anti-apoptosis and anti-oxidation remained the same. The topic is interesting. The following comments should be addressed before the paper can be considered for publication

1� In the abstract, the authors mentioned that “intermittent hypoxia preconditioning leads to differences in the secretion of certain cytokines”. The authors should specify which cytokines were involved and indicate whether their levels increased or decreased.

2� In the introduction, the authors should introduce intermittent hypoxia preconditioning and discuss whether this condition has been studied using mesenchymal stem cells.

3� The authors should do some tests to make sure that the isolated ADSCs meet the minimum MSC criteria recommended by ISCT. First, they must adhere to plastic in standard culture conditions. Second, they must express a specific set of surface markers, namely CD105, CD73, and CD90, while lacking markers like CD45, CD34, CD14, CD11b, CD79a, CD19, and HLA-DR. Third, they must be able to differentiate into osteoblasts, adipocytes, and chondrocytes in vitro.

4� The authors should also study the effects of intermittent hypoxia preconditioning on proliferation and stemness of ADSC.

5� The authors should propose some next steps or future studies.

6. PLOS authors have the option to publish the peer review history of their article (what does this mean? ). If published, this will include your full peer review and any attached files.

**Do you want your identity to be public for this peer review?** For information about this choice, including consent withdrawal, please see our Privacy Policy .

Reviewer #1: No

Reviewer #2: No

---

## [Author Response · Author response to Decision Letter 1]

15 Dec 2025

We have provided responses in both the cover letter and the Response to Reviewers document.

---

## [Editor Report · Decision Letter 1]

18 Dec 2025

Effects of intermittent hypoxia on adipose-derived mesenchymal stem cells in protecting alveolar type II cells from injury

PONE-D-25-45087R1

Dear Dr. Li,

We’re pleased to inform you that your manuscript has been judged scientifically suitable for publication and will be formally accepted for publication once it meets all outstanding technical requirements.

Kind regards,

Cheng-Maw Ho

Academic Editor

PLOS One
---

## [Editor Report · Acceptance letter]

3 Nov 2025

PONE-D-25-45087R1

PLOS One

Dear Dr. Li,

I'm pleased to inform you that your manuscript has been deemed suitable for publication in PLOS One. Congratulations! Your manuscript is now being handed over to our production team.

Kind regards,

on behalf of

Dr. Cheng-Maw Ho

Academic Editor

PLOS One